# Are Medication Swallowing Lubricants Suitable for Use in Dysphagia? Consistency, Viscosity, Texture, and Application of the International Dysphagia Diet Standardization Initiative (IDDSI) Framework

**DOI:** 10.3390/pharmaceutics12100924

**Published:** 2020-09-28

**Authors:** Marwa A. Malouh, Julie A.Y. Cichero, Yady J. Manrique, Lucia Crino, Esther T. L. Lau, Lisa M. Nissen, Kathryn J. Steadman

**Affiliations:** 1School of Pharmacy, The University of Queensland, Brisbane, QLD 4072, Australia; m.abumalouh@uq.edu.au (M.A.M.); j.cichero@uq.edu.au (J.A.Y.C.); yadyjuliana.manriquetorres@qut.edu.au (Y.J.M.); lucia.crino@gmail.com (L.C.); et.lau@qut.edu.au (E.T.L.L.); l.nissen@qut.edu.au (L.M.N.); 2School of Clinical Sciences, Queensland University of Technology, Brisbane, QLD 4001, Australia

**Keywords:** rheology, yield stress, Bostwick consistometer, xanthan gum, carrageenan, texture analysis

## Abstract

Medication lubricants are thick liquids or gels that are designed to aid swallowing of solid oral dosage forms. Tablets and capsules are placed within a spoonful of the product for swallowing. The aim of this study was to describe and compare commercially available medication lubricants in terms of textural suitability for patients with dysphagia. Twelve medication lubricants were characterised according to the International Dysphagia Diet Standardisation Initiative (IDDSI) framework. Apparent viscosity, yield stress, thickness consistency, and various texture features were compared. Gloup Forte was the only medication lubricant classified as IDDSI level 4 (pureed/extremely thick) at room (24 °C) temperature. Four other Gloup products were IDDSI level 3 (liquidised/moderately thick) at room temperature but testing at 4 °C or pouring from the container instead of using the pump dispenser resulted in classification as IDDSI level 4. The IDDSI Flow test would have classified MediSpend and Slo Tablets as IDDSI level 3, but their very low yield stress led to these fluids flowing too quickly through the prongs of a fork and so these were classified as <3. Severo was IDDSI level 2. Heyaxon and the two versions of Magic Jelly tested contained lumps, and Swallow Aid had exceptionally high viscosity, hardness, adhesiveness, and gumminess, classifying them as IDDSI Level 7 (“regular textures”) and therefore as unsuitable for people with dysphagia according to IDDSI. This study provides valuable information to help with the selection of a safe medication lubricant with appropriate thickness level suited to each individual with dysphagia.

## 1. Introduction

Solid oral dose forms may be a simple, convenient, non-invasive and economically efficient approach to medication delivery, but some individuals find it extremely difficult to swallow them whole. A recent report on swallowing problems associated with dietary supplements, specifically multivitamins or calcium supplements, found that choking (86%) was the most frequently reported swallowing problem [1]. Struggling to swallow solid oral medication may eventually drive patients to stop taking some or all of what they have been prescribed [2]. One group who find it challenging to swallow solid oral medications are those with dysphagia. These people face problems with swallowing food and drinks safely without aspirating them, so it can be dangerous to attempt to swallow tablets and capsules whole [3,4]. Dysphagia is associated with age-related physiological changes, and a wide variety of neurological, muscular, respiratory disorders, e.g., stroke, Parkinson’s disease, metabolic myopathy [5,6]. Dysphagia is a highly prevalent medical condition that is estimated to affect around 30–60% of nursing home residents, 10–30% of hospitalised elderly patients, and at least 35% of stroke patients [7,8].

Consuming fluids with the right thickness and texture properties is vital for patients with dysphagia. Fluids with an inappropriate thickness might expose patients to serious health consequences. For example, a fluid that is too thin may cause aspiration, while over-thickened fluids increase the risk of post-deglutition oropharyngeal residue and choking [9]. Texture properties such as stickiness or slipperiness, more often associated with food texture, are also important considerations for thickened liquids. In clinical practice, patients that are identified to have a swallowing disorder (e.g., after acute stroke) are referred to speech and language pathologists for evaluation and management. For each individual, the severity of dysphagia is assessed, and the safe food texture and fluid thickness level that meet the patient’s needs are determined [10,11]. The severity of swallowing impairment in patients with dysphagia is varied, with some patients able to safely swallow thin liquids while concurrently experiencing difficulty with managing mixed consistencies, while others are unable to even safely swallow their own saliva [12]. In managing dysphagia, the thickness of fluids must be available in various levels to meet the evolution of the condition, as each patient’s needs differ depending on the diagnosis and progression of dysphagia. While the focus has been on liquid thickness, the evaluation of lubricants to assist swallowing of medications for the first time highlights the importance of considering other bolus properties such as stickiness or adhesiveness that could cause the bolus to stick to the mucosa of the mouth or in the pharynx, increasing aspiration risk and swallowing dysfunction. A bolus that is very thick and very sticky provides increased risk for people with dysphagia.

The International Dysphagia Diet Standardization Initiative (IDDSI) recently published a global guideline to identify thickness levels of texture-modified foods and drinks for patients with dysphagia in order to address the lack of comparability between the various national guidelines that exist [13,14]. The framework enables classification of nutritional products on a continuum of 8 levels (0–7) in which drinks are described by levels 0–4 and foods are covered by levels 3–7. This international guideline replaces the different national standards, aiming to improve patient safety; facilitate communication within and between healthcare providers and patients; and to provide simple, easy, and applicable measurement methods suitable for use in different settings and nations exist [13]. Evaluation of the effectiveness of the IDDSI framework concludes that it can be an efficient tool for clinical application [15]. This guideline has been broadly welcomed by food manufacturers, stakeholders and aged care facilities, and it is being applied in many countries [16,17].

In order to facilitate swallowing of solid oral medications, patients with dysphagia often crush tablets and open capsules and mix the powder with a thick vehicle such as naturally thick food such as pudding, jam, yoghurt, or fluids thickened with a commercial polysaccharide product [18,19]. The viscosity of these food and fluids slows oral transit time, which in turn reduces the risk of aspiration, making them safer to swallow [20]. A relatively recent group of products to be commercialised are lubricant gels or fluids that are designed to help with swallowing whole solid oral dosage forms. A tablet or capsule is placed within a spoonful of the medication lubricant for swallowing. These products are generally designed for use by anyone that finds it difficult to swallow whole tablets/capsules with water, for whom the thickness, texture, and flavour can help to disguise the presence of a medication. For most people, the choice between these products comes down to personal preference. However, some of these products are labelled as being suitable for use by patients who have dysphagia but data supporting this use is not available. Various techniques have been employed to measure and describe dysphagia-oriented diets for dysphagia management. Rheological properties (i.e., viscosity and yield stress) provide important information that have a profound effect on swallowing safety [21], and although there are no standard conditions prescribed for rheological testing, a shear rate of 50 s^−1^ is commonly used as an indicator of the shear rates of swallowing, which facilitates comparison between studies [15,22]. A Bostwick consistometer has been used to assess the consistency of dysphagia-oriented thickeners by several manufacturers, as it has good reproducibility and offers accessible measurements [23]. Texture features, such as hardness and adhesiveness, are also characteristics that are considered in some studies because these affect the swallowing process [24]. Although several guidelines for the classification of dysphagia-oriented food and drinks based on thickness level and texture features have been developed, these have not been widely or consistently applied [25].

This study aimed to describe and compare commercially available medication swallowing lubricants in terms of thickness and textural suitability for being consumed by patients with dysphagia. People who have trouble swallowing solid doses (tablets and capsules) often use naturally thick foods such as pudding, yoghurt or fluids thickened with a commercial polysaccharide, so commercially available medication lubricants should possess similar textural properties. The medication lubricants were classified according to the IDDSI framework, and rheological properties (viscosity, yield stress), thickness consistency (the flow distance in the Bostwick consistometer), and texture features (hardness, adhesiveness, cohesive force, gumminess, cohesiveness, springiness, and consistency) were measured. As the rheological behaviour and the thickness level of liquids depend on the conditions used for testing [26,27], all medication lubricants were tested at two serving temperatures typically used in practice, room (24 °C) and fridge (4 °C). Additionally, some products that are designed to be administered using a pump pack provided in the packaging were compared when pumped versus poured from the bottle to investigate whether shearing forces exerted by the pump affect lubricant characteristics.

## 2. Materials and Methods

### 2.1. Materials

A total of 12 commercial medical devices were tested: five different Gloup products, Heyaxon, MediSpend, two products from the Magic Jelly range, Severo Swallowing Gel, Slo Tablets, and Swallow Aid (Table 1). All 12 products are designed as medication swallowing aids for individuals who struggle to swallow their oral solid medication whole. All are presented as ready to use products that are placed onto a spoon, and the tablet/capsule is placed into it before swallowing. Note that Gloup is marketed under the name Assure Slide and Phazix in the USA, and Med-Easy (Fagron, Newcastle, UK) is the same product as MediSpend (Fagron, Rotterdam, The Netherlands) according to our assessments, so these products were not tested in this study.

### 2.2. Methods

The medication lubricants were classified according to the IDDSI framework and characterised for pH, density, rheological properties (viscosity, yield stress), Bostwick thickness consistency, and texture properties. The effect of temperature was tested by storing all lubricants in an air-conditioned laboratory at approximately 24 °C and fridge (4 °C) temperature, for 24 h before assessment. Based on advice given the manufacturer of Gloup that the product can be retrieved by using the pump provided with the packaging or by pouring it from the bottle, these retrieval methods were compared for the five Gloup products.

#### 2.2.1. pH and Density

pH was measured using a S220 SevenCompact pH meter (Mettler-Toledo, Port Melbourne, Australia). Density was measured by pre-calibrated pycnometer at room temperature.

#### 2.2.2. The International Dysphagia Diet Standardization Initiative (IDDSI)

The IDDSI guidelines were followed to classify the thickness of the medication lubricants, working stepwise through the IDDSI Flow test, Fork Drip and Spoon Tilt tests [28]. The IDDSI framework consists of eight levels of thickness, levels 0 to 4 describe the thickness level of fluids, while levels 3 to 7 describe the texture of food. Measurements were made at room temperature (24 °C). Samples were stored at room temperature or in the fridge (4 °C) and taken from their storage temperature immediately prior to each test. All IDDSI assessments were applied to three replicate samples of each medication lubricant.

*IDDSI Flow test*: A 10 mL central luer slip tip syringe (BD, Singapore REF 302143) was used in accordance with the IDDSI syringe specifications. The syringe was filled up to the 10 mL mark with a sample of the test product using another 10 mL syringe, and then, the sample was allowed to flow freely for 10 s using a timer. Based on the remaining volume of the sample after 10 s of flow, the IDDSI level was determined as level 3 (no less than 8 mL remaining), level 2 (4–8 mL remaining), level 1 (1–4 mL remaining) or level 0 (less than 1 mL remaining).

*Fork Drip test*: Fluids at level 3 or 4 according to the IDDSI Flow test were assessed using the Fork Drip test. A 4-prong metal dinner fork with 4 mm space between the tines of the fork was dipped into and then pulled up through a sample of the test fluid and observed. The IDDSI level was determined by visual assessment; samples that remained as a mound above the fork were classified as level 4, while samples that dripped in slow dollops through the fork prongs were classified as level 3. Samples that ran straight through the prongs of the fork were considered not to have the characteristics of a true level 3 fluid and so were classified as <3.

*Spoon Tilt test*: Fluids at level 4 according to the Fork Drip test were assessed using the Spoon Tilt test. A tablespoon of the sample was observed for its ability to hold its shape on the spoon and then for its behaviour as the spoon was tilted to dislodge the sample. The sample passed the test if it could hold its shape on the spoon and could be easily dislodged when the spoon was tilted leaving little residue or stickiness.

Fluids that were not valid to be tested using IDDSI flow test, for example those with a mixed texture or containing lumps, and fluids that failed to meet IDDSI level 4 criteria according to the Fork Drip and Spoon Tilt tests, were classified as IDDSI level 7 Regular in accordance with the IDDSI Framework detailed definitions.

#### 2.2.3. Rheological Properties

Viscosity and yield stress measurements of medication lubricants were performed using a stress-controlled rheometer HR-3 (TA instruments c/o Waters Australia Pty. Ltd., Sydney, Australia) with parallel plates. All samples were retrieved from room or fridge temperature storage immediately prior to testing, but then allowed to equilibrate at the initial temperature (24 °C) for 300 s to rest the sample within the plates before the steady-shear rheological measurements were started. Viscosity measurements involved the peak hold test at 24 °C with a shear rate of 50 s^−1^ over a period of 60 s, with a gap from 200 to 500 μm depending on the gap suitability for the sample. This shear rate is commonly used to evaluate liquid foods as an indicator of the shear rates of swallowing, and using this value facilitates comparison between studies [15,22]. Yield stress was determined using a flow sweep test by applying a wide range of stresses (0.001–1000 Pa) at 24 °C. All rheological measurements were applied to three replicate samples of each medication lubricant.

#### 2.2.4. Bostwick Thickness Consistency

Each medication lubricant was tested using the Bostwick consistometer (CSC scientific company, Fairfax, VA, USA) one sample at a time. The reservoir of the Bostwick consistometer was filled with sample (75 mL), and the top was levelled off using a spatula. The trigger was pressed down to open the gate of the reservoir, and at the same time, a stopwatch was started. The sample was then free to flow through the slightly inclined trough. A reading of the distance (in mm) that each sample flowed was taken after 30 s. Measurements were made at room temperature (24 °C). Samples were stored at room temperature or in the fridge (4 °C) and taken from their storage temperature immediately prior to each test. Measurements were carried out for three replicate samples of each medication lubricant.

#### 2.2.5. Texture Properties

A Brookfield Texture Analyser CT3 (Ametek Brookfield, Middleboro, MA, USA) equipped with 4500 g load cell and texture analysis software program, was used to determine the texture properties of the medication lubricants. Texture profile analysis and extrusion tests were performed, with each test applied to three replicate samples of each medication lubricant [29,30]. Measurements were made at room temperature (24 °C). Samples were stored at room temperature or in the fridge (4 °C) and taken from their storage temperature immediately prior to each test. Texture profile analysis (TPA) was performed using a clear cylinder probe of 12.7 mm diameter and 35 mm length (TA10). Approximately 27 g of the medication lubricant was filled in a standard beaker (40 mL). The sample was compressed at pretest, test and return speeds of 2, 1, and 1 mm s^−1^ to a distance of 20 mm. A force of 0.1 g was used as a trigger value. The textural features measured were hardness, adhesiveness, cohesiveness, gumminess, springiness. Back and forward extrusion tests were performed using an extrusion cell (TA-DEC), consisting of a cylinder vessel of 40 mm diameter and 50 mm depth. For the back-extrusion test, a closed disc was positioned in the base of the vessel. A constant amount of medication lubricant was poured into the extrusion cell. A compression plunger with 38 mm diameter was programmed to compress the samples at a speed of 2 mm s^−1^ and allow flow upward through the space between the plunger and the extrusion cell container. A two-cycle compression test was performed with 20 mm target distance and 200 g trigger load to measure consistency, hardness, adhesive force, and adhesiveness. For the forward extrusion test, a disc with one 2 mm diameter hole was positioned in the base of the extrusion vessel. The two-cycle compression test was performed with a 15 mm target distance, 200 g trigger load and compression plunger with 39.9 mm diameter to measure hardness and consistency.

### 2.3. Statistical Analysis

All statistical analyses were performed using Prism version 8.00 (GraphPad Software, San Diego, CA, USA). Results are given as mean and standard error. One-way ANOVA followed by a Bonferroni multiple comparisons test was used to determine whether there were differences in properties between the medication lubricants. Comparison of room and fridge temperature and pour versus pump for properties of the Gloup products were investigated with two-way ANOVA followed by a Bonferroni multiple comparisons test. Correlations between viscosity and thickness consistency and yield stress were investigated by calculating Pearson’s correlation coefficient (r). Differences were considered significant at *p* < 0.05.

## 3. Results

### 3.1. The International Dysphagia Diet Standardization Initiative (IDDSI)

At room temperature (24 °C) Gloup Forte was the only lubricant classified as IDDSI level 4 (Table 2). Its shape held and did not fall through the prongs in the Fork Drip test, and it slid easily off as a single dollop in the Spoon Tilt test without leaving a sticky residue. All other Gloup products were classified as IDDSI level 3 because although they barely flowed through the syringe in the IDDSI Flow test, they dripped slowly through the prongs in the Fork test (Table 2). MediSpend and Slo Tablets flowed slowly through the syringe leaving 8.7–9.1 mL but flowed continuously through prongs of a fork and so did not behave as required of an IDDSI level 3 fluid; consequently, we classified these products as <3 as a way of indicating that they did not meet the classification of either IDDSI 2 or 3. Severo was classified as IDDSI level 2 based on the IDDSI Flow test as the volume remaining in the syringe after 10 s was 7.5 mL (Table 2).

Storage and testing at 4 °C resulted in Gloup products thickening enough for their classification to increase to level 4 because of a change in their response to the Fork Drip test, but it did not change the classification of the other medication lubricants (Table 3). The classification of Gloup products also increased to level 4 if they were poured out of the bottle instead of pumped out using the pump dispenser provided with the packaging, due again to holding their form and not dripping through the prongs in the Fork Drip test (Table 4).

Fluids that contain lumps, fibres, gristle, bone, husks, or shell are not valid to be tested using the IDDSI Flow test, as these contents increase choking risk when consumed by patients with dysphagia. Both types of Magic Jelly (dysphagia and adult) and Heyaxon contained lumps; consequently, these were classified as IDDSI level 7 (Table 2). Swallow Aid showed high stickiness as it did not dislodge from the spoon during the spoon tilt test, so it failed the suitability criteria for IDDSI level 4 and was therefore classified as IDDSI level 7 (Table 2).

### 3.2. Rheological Properties

Swallow Aid had exceptionally high viscosity (51 Pa.s at room temperature) compared with all other medication lubricants (Figure 1A), with the next highest being Gloup Forte (2.1 Pa.s at room temperature). Other Gloup products recorded viscosity readings ranging between 0.64–0.9 Pa.s at room temperature, while Severo, Medispend, Heyaxon, Magic Jelly and Slo Tablets had the lowest viscosity (0.35–0.47 Pa.s). Storage temperature had a minor role on the viscosity, which in this study involved samples being equilibrated and tested in the rheometer at 24 °C. When all the products were included in the comparison, measurements on products stored at room temperature were not significantly different from products stored in the fridge (Figure 1A). However, if only the Gloup products were considered, there was a significant effect on viscosity (*p* < 0.05), with products from the fridge being between 0.01 and 0.16 Pa.s thicker than room temperature (Figure 1A). Pouring the Gloup products directly from the container (data not shown) instead of using the pump dispenser (Figure 1A) also resulted in a higher viscosity reading of between 0.10–0.21 Pa.s at room temperature (*p* < 0.001).

In terms of yield stress, Gloup Forte was the highest value measured at 38 Pa (Figure 2). Yield stress for the other products varied between 12.5 Pa for Gloup Low Sugar down to 1.6 Pa for Slo Tablets. There was a strong positive correlation between yield stress (Figure 2) and viscosity (Figure 1A) across the products tested at room (r = 0.951) temperature. However, it was not possible to achieve a reading for Swallow Aid or Severo using the geometry applied to the other medication lubricants.

### 3.3. Bostwick Consistometer

With Bostwick consistometer measurements, a greater flow distance indicates lower thickness consistency (Figure 1B). For the Gloup products, Gloup Forte travelled the least (2.1 and 3.3 cm for fridge and room temperature) and Gloup Original strawberry/banana travelled the furthest (6.6 and 8.4 cm). There was strong negative correlation between flow distance (Figure 1B) and viscosity (Figure 1A) across the Gloup products tested at room (r = −0.924) and fridge (r = −0.940) temperatures. In line with this, Swallow Aid travelled the shortest distance in 30 s (2.9 cm) and had the greatest viscosity (51 Pa.s) at room temperature. However, while the other products generally had larger flow distances (6.5–22 cm) and lower viscosities (0.37–0.45 Pa.s), the overall correlation between flow distance and viscosity was not as strong when calculated across all 12 products (room temperature r = −0.568, fridge temperature r = −0.579).

Cold temperature increased thickness consistency of most of the medication lubricants, as it decreased the distance that the lubricants flowed in the Bostwick consistometer (*p* < 0.0001) (Figure 1B); temperature was a significant main effect (*p* < 0.0001) according to ANOVA. Additionally, pouring Gloup samples (data not shown) instead of using the pump dispenser (Figure 1B) resulted in significant (*p* < 0.0001) reductions in flow distances of between 0 and 1.3 cm at both serving temperatures.

### 3.4. Texture Characterisation

Swallow Aid had significantly larger values than all other products for texture properties of hardness, adhesiveness, and gumminess but was no different in cohesiveness and springiness (Table 5). It was too thick for testing with the extrusion test, which is used for fluids and semisolid fluids [31].

Gloup Forte was approximately twice as hard and gummy than the other medication lubricants, but no different in terms of adhesiveness, cohesiveness, and springiness (Table 5). The significant difference of Gloup Forte to all others was confirmed by the extrusion tests, as it provided around two to five times higher values of hardness, consistency, cohesiveness and cohesive force during back extrusion (Table 6). It was also among the most difficult to force through a small hole in the forward extrusion test, matched or exceeded only by the medication lubricants that contained lumps (Heyaxon and Magic Jelly).

There was very little difference between any of the other medication lubricants in any of the characteristics measured by texture profile analysis (Table 5). Testing at cold temperature caused a small reduction in texture profile values (Table 5), particularly for Gloup products for which cohesiveness and springiness reduced and hardness increased significantly (*p* < 0.01). Pouring the Gloup products instead of using the pump had little impact, with only adhesiveness exhibiting a small but significant reduction (data not shown).

The extrusion tests differentiated between the Gloup products and the other smooth fluid medication lubricants (Slo Tablets, Medispend, Severo). The Gloup products all had significantly greater values for parameters measured by back extrusion (hardness, consistency, adhesive force, and adhesiveness) and forward extrusion (hardness, consistency) than Slo Tablets, Medispend and Severo (Table 6).

## 4. Discussion

The thickest level for liquids in IDDSI classification guideline is IDDSI level 4. IDDSI level 4 encompasses products that may have previously been described by terms such as extremely thick, full thickness, spoon thick or pudding thickness [32]. They do not drip through a syringe, hold their form on a fork, and slide off a spoon without leaving a sticky residue. IDDSI Level 4 products are appropriate for patients with dysphagia that cannot safely drink liquids through a straw and require them to be thickened enough to consume from a spoon [17]. This consistency can improve safety by reducing the potential for liquid aspiration and reducing bolus velocity in patients with dysphagia [9,33,34]. Gloup Forte was the only medication lubricant that clearly behaved as IDDSI level 4 at both room (24 °C) and fridge (4 °C) serving temperatures. Gloup Forte has significantly higher viscosity, yield stress, and thickness consistency compared with other lubricants. Indeed, the viscosity of Gloup Forte (2.1 Pa.s) was consistent with other IDDSI level 4 dysphagia-oriented products, which have viscosity at 50 s^−1^ in the range 0.64 to 3.78 Pa.s [35].

IDDSI level 3 is a “moderately thick” liquid, encompassing products previously described using terms such as half thickness or honey thick [32]. Level 3 liquids leave more than 8 mL liquid in an IDDSI approved 10 mL syringe following 10 s of flow and drip slowly through the prongs of a standard dinner fork such that it is not possible to consume them using a fork. When served at room temperature, four Gloup products were classified as IDDSI level 3 because they met these requirements. They were clearly at the thicker end of the level 3 spectrum, as it only took a reduction in temperature or removal of the pump dispenser to switch them to level 4 classification. Indeed, the room temperature Gloup measurements for viscosity (0.6–0.9 Pa.s) were more similar to dysphagia-oriented products that classify as IDDSI level 4 (0.64 to 3.78 Pa.s) than level 3 (0.25 to 0.6 Pa.s) [35]. Additionally, Gloup products flowed in a Bostwick consistometer (5–8 cm in 30 s) at a rate that was more similar to IDDSI level 4 dysphagia-oriented products (7.3 to 8.6 cm) than level 3 (8.3 to 18.9 cm) [23].

We classified MediSpend and Slo Tablets as <3 in the IDDSI tests. These products would be classified as IDDSI level 3 using the IDDSI Flow test, as 8 mL or more of the liquid was retained after 10 s of flow. Traditional rheometry measurements of the samples used in this study provided their viscosity as 0.45–0.42 Pa.s, which aligns with the range for other IDDSI level 3 dysphagia-oriented products (0.25 to 0.6 Pa.s) [35]. Assessment of MediSpend and Slo Tablets using the Bostwick consistometer (15–17 cm in 30 s) was at the faster end of the flow range (8.3 to 18.9 cm in 30 s) measured for dysphagia-oriented liquids that classify as IDDSI level 3 according to the IDDSI Flow test [23]. However, their behaviour in the Fork Drip test did not correspond with the requirements for IDDSI level 3 because they were not retained at all by a fork and instead flowed continuously through the prongs. Hence, in accordance with the IDDSI detailed descriptors [28], these products are not suitable to be consumed by patients with dysphagia who require liquids to be thickened to IDDSI level 3 for safe swallowing.

Severo was the only product that was classified as IDDSI level 2 (mildly thick liquid) according to the IDDSI flow test because less than 8 mL of fluid remained in the syringe after 10 s of flow. In fact Severo was the same as Slo Tablets and Medispend in all of the textural features measured, and viscosity (0.47 Pa.s) was also in the range measured for IDDSI level 3 dysphagia-oriented products (0.25 to 0.6 Pa.s) rather than level 2 (0.08 to 0.2 Pa.s) [35]. However, it flowed quickly in the Bostwick consistometer (21.1 cm in 30 s), well within the range measured for level 2 dysphagia-oriented products (16.7 to 22.2 cm) [23]. In a swallowing study involving 120 patients with oropharyngeal dysphagia, only 25 safely completed a series of three swallows of a thin liquid (5 mL, 10 mL, 20 mL), with other 95 patients exhibiting voice changes, cough or oxygen desaturation [9]. Hence, while products at IDDSI level 2 present no problem to healthy people [9] and may be a useful aid in oral medication delivery, only a small proportion of dysphagic patients are likely to be able to safely swallow thin fluids and evaluation of swallowing function prior to use is essential.

Temperature can profoundly affect the properties of materials, especially hydrocolloids [36]. For example, the viscosity of thickened juices significantly increased at cold temperature [37]. The effect of temperature can be described by the Arrhenius relationship, which illustrates the inverse relationship between temperature and viscosity, as lower temperature leads to thicker fluids [38]. The thickening agents used in each product determine the extent to which temperature affected viscosity. Carrageenan solutions are highly affected by temperature, exhibiting a significant increase in viscosity at low temperature when compared with xanthan gum and starch-based solutions [39]. Indeed, there was a small but significant effect of temperature on viscosity of Gloup products, which contain carrageenan, along with slower flow in the Bostwick consistometer and an associated change in IDDSI category. Storage and testing at fridge temperature did result in slower flow in the Bostwick consistometer for the starch-based (Slo Tablets and MediSpend) and cellulose (Severo) products, but their IDDSI classification was unchanged.

In addition to temperature, shearing and pressure affect the properties of materials [36]. In our tests, four Gloup products were classified as IDDSI level 3 for samples taken using the pump dispenser, but IDDSI level 4 if the dispenser was removed and the products were poured from their bottles. We take this to indicate the effect of shearing forces exerted by the pump dispenser. Using the pump dispenser resulted in small but significant decrease in viscosity and thickness consistency, in comparison to when the material was poured. This was associated with a difference in behaviour in the Fork Drip test such that the poured samples retained their form and did not drip while the pumped samples dripped very slowly through the prongs.

Yield stress provides valuable information about the effort needed to swallow a bolus and is strongly aligned with the results of the fork test [22]. Gloup Forte (IDDSI level 4) was able to hold its shape on a fork due the combination of high viscosity (2.1 Pa.s) and very high yield stress (38 Pa) characteristics. The Gloup products at the thicker end of level 3 retained some form on a fork with slow dripping through the prongs, which was associated with relatively high viscosity for a level 3 (0.6–0.9 Pa.s) and yield stress between 4.9–12.5 Pa. The viscosity and IDDSI flow test would indicate Slo Tablets and Medispend might be level 3; however, these products both flowed straight through the prongs of a fork. We believe that this property is associated with the noticeably lower yield stress (1.6–2.5 Pa). It is difficult to directly compare yield stress values in the literature because the method of measurement affects the outcome; however, we are confident that the yield stress values we measured for these products were certainly more characteristic of IDDSI level 2 fluids (nectar/quarter/mildly thick) than level 3 fluids (honey/half/moderately thick) measured in other studies [23,40,41]. We were unable to measure a yield stress for the IDDSI level 2 product, Severo, which is not unusual for materials thickened with cellulose [42].

Some of the products that were tested were found to be inappropriate for use by patients with dysphagia. A number of properties of Swallow Aid make it unsuitable for use in dysphagia. It had exceptionally high viscosity (51 Pa.s), which was reflected in the significantly higher hardness (16.5 g) and the shortest flow distance (2.9 cm in 30 s) when compared with other products. Furthermore, the product did not yield within the range of stresses tested in this study. It was too thick to flow through the IDDSI Flow test, but it failed classification as level 4 when it stuck to the spoon in the IDDSI Spoon Tilt test deeming it too sticky. This corresponds with the measurements of adhesiveness (1.95 mJ) and gumminess (16.4 g) in the texture profile analysis that were significantly higher than other products. Although evidence shows that increasing the thickness level of fluids is associated with a decreased risk of aspiration, consuming over-thickened fluids might increase post swallowing residue, which in turn increases the risk of aspiration [9]. Additionally, a bolus with high adhesiveness and gumminess tends to stick to the inner part of the mouth, also increasing the risk of aspiration [24,25,43]. The texture profile analysis and the IDDSI Spoon Tilt test are valuable measures in this regard.

Products containing a mixed texture are generally viewed as inappropriate for use by people with dysphagia because they can complicate the swallowing process and therefore increase the risk of aspiration [12,13]. Heyaxon and both types of Magic Jelly tested here (“for adults” and “dysphagia”) contained lumps of jelly within a thin continuous phase. These products had viscosity (0.35–0.38 Pa.s), thickness consistency (8.2–10.8 cm in 30 s), yield stress (4–9 Pa), and texture features that were similar to products within IDDSI level 3, which reflect properties of the continuous phase. In contrast, they showed extremely high hardness and consistency values in the forward extrusion test, in keeping with the lumps getting stuck in the aperture of the extrusion vessel. Therefore, these products were deemed invalid and inappropriate for people with dysphagia according to the IDDSI framework.

## 5. Conclusions

Commercially available products that are designed to assist people to swallow solid dose medications vary considerably in viscosity, yield stress, flow, and texture properties. Any of these commercial products may be useful for those who simply have an aversion to swallowing tablets and capsules whole or find it difficult to achieve using water, and the thickness and texture properties are of no concern for these people. However, these products are not necessarily safe for use by all patients with dysphagia, especially for those who require careful adjustment of the texture and consistency of foods and fluids to minimise the risk of aspiration. The IDDSI framework is usually used to classify the suitability of food and drink products for dysphagia, and here we have applied it to medication lubricants. These products are a highly diverse group, comprised of a variety of thickeners and thicknesses. Some products that are commercially available as medication swallowing aids are not considered to be appropriate for use by patients with dysphagia: Swallow Aid is exceptionally thick and sticky and did not yield and flow in the tests applied in this study, so there are concerns about the risk of choking and aspiration if used by patients with dysphagia; Magic Jelly and Heyaxon contain lumps of jelly, and mixed textures are considered to be inappropriate for patients with dysphagia according to the IDDSI framework. An important consideration will be the effect that mixing tablets/capsules, whether whole, split, or crushed, into the thick vehicle before administration affects the vehicle’s properties. This is an issue that has not yet been considered for any commercial or food product that is used for medication delivery.

Gloup Forte is the only product tested that is firmly within the IDDSI level 4 category, meeting the IDDSI testing requirements and rheologically having high viscosity together with high yield stress values, without being too hard, too adhesive, or too gummy. The other Gloup products (Original, Sugar Free or Low Sugar) are at the thicker end of the IDDSI level 3 range with normal use and have lower yield stress values than Gloup Forte. These products behaved more like an IDDSI level 4 product if used cold or poured from the bottle, so it is recommended that users test these products under their own conditions of use. The thinner products were classified as IDDSI < 3 (Slo Tablets, MediSpend) or level 2 (Severo), were much thinner products than Gloup, with lower viscosity, yield stress, and considerably faster flow. The findings from this study are valuable for helping healthcare professionals select a medication lubricant that will be safe for their patients with dysphagia to swallow.

## Figures and Tables

**Figure 1 pharmaceutics-12-00924-f001:**
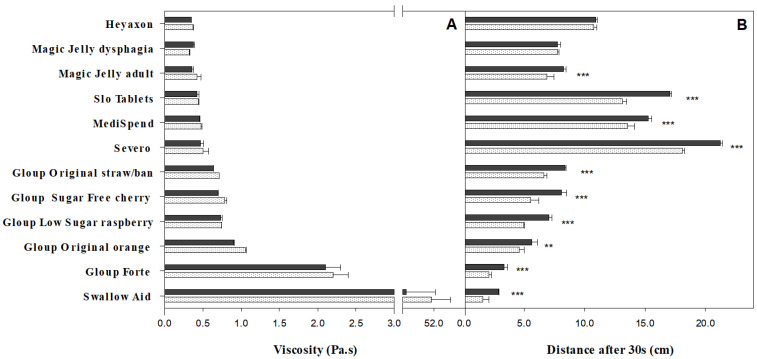
Viscosity and thickness consistency for medication lubricants served at room temperature, 24 °C (black bars), or fridge temperature, 4 °C (white with pattern bars). (**A**) Viscosity measured at 50 s^−1^ (Pa.s) and (**B**) Bostwick thickness consistency measured as distance (cm) moved in 30 s. Statistically significant differences between measurements at room and fridge temperature are indicated (** *p* < 0.01. *** *p* < 0.001). The bars indicate mean ± se for three replicates.

**Figure 2 pharmaceutics-12-00924-f002:**
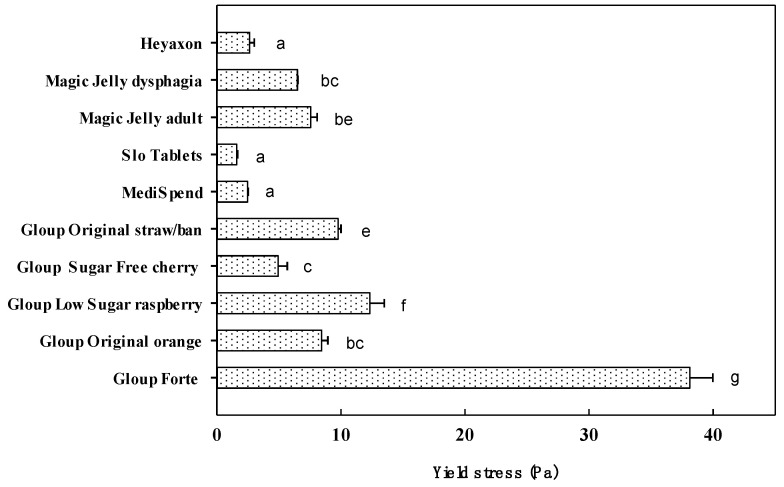
Yield stress values for medication lubricants measured across stress range 0.001–1000 Pa, at room temperature (24 °C). Significant difference (*p* < 0.05) between mean values is indicated by unlike letters (**a**–**f**) above the bar.

**Table 1 pharmaceutics-12-00924-t001:** Details of the twelve medication swallowing lubricants tested in this study. Flavour and composition details were taken from the product container; pH and density were measured at room temperature (24 °C).

Product Name	Manufacturer	Flavour	Composition	pH	Density (kg/m^3^)
Gloup Forte	Rushwood, Raamsdonksveer, The Netherlands	vanilla	Water, dried glucose syrup, sucrose, carrageenan, maltodextrin, potassium sorbate, citric acid, aroma	5.16	1.1
Gloup Low Sugar	Rushwood, Raamsdonksveer, The Netherlands	raspberry	Water, xylitol, carrageenan, maltodextrin, potassium sorbate, citric acid, colour, aroma	5.36	1.04
Gloup Original	Rushwood, Raamsdonksveer, The Netherlands	orange	Water, carrageenan, maltodextrin, potassium sorbate, sucrose, calcium chloride, citric acid, colour, aroma	5.11	1.03
Gloup Original	Rushwood, Raamsdonksveer, The Netherlands	strawberry/banana	Water, carrageenan, maltodextrin, potassium sorbate, sucrose, calcium chloride, citric acid, colour, aroma	5.19	1.05
Gloup Sugar Free	Rushwood, Raamsdonksveer, The Netherlands	cherry	Water, carrageenan, maltodextrin, potassium sorbate, aspartame, calcium chloride, citric acid, (natural) colour, (natural) aroma	5.19	1.02
Heyaxon	Jian An Pharmaceutical, Shenzhen, China	peach	Water, erythritol, xylitol, agar, citric acid, xanthan gum, sodium citrate, locust bean gum, pigment, peach perfumes, sucralose, crocine, glycerol fonostearate	3.77	1.03
Magic Jelly (adult)	Ryukakusan, Tokyo, Japan	lemon	Erythritol, hydrogenated maltose starch syrup, agar, gelling agents (polysaccharide thickeners, calcium acetate, sweetener (Stevia)	3.65	1.04
Magic Jelly (dysphagia)	Ryukakusan, Tokyo, Japan	lemon	Erythritol, hydrogenated maltose starch syrup, agar, gelling agents (polysaccharide thickeners, calcium acetate, sweetener (Stevia)	3.64	1.05
MediSpend	Fagron, Rotterdam, The Netherlands	lemon	Purified water, modified food starch, natural lemon flavour, sodium citrate, citric acid, sucralose, sodium benzoate	4.2	1.01
Severo	IMS Medical, Grootebroek, The Netherlands	anise	purified water, cellulose gum, flavour (anise), citric acid, potassium sorbate, aspartame, acesulfame K	4.41	1
Slo Tablets	Slo Drinks, Glossop, UK	cherry	Purified water, modified food starch, cherry flavour, sodium citrate, citric acid, sucralose, sodium benzoate, malic acid, simethicone	4.28	1.01
Swallow Aid	National Consumer Products Inc, USA	cherry	Malitol, Glycerine, Carboxymethyl cellulose, Acesulfame Potassium	-	-

- not available.

**Table 2 pharmaceutics-12-00924-t002:** Medication lubricant classification according to IDDSI testing methods at room temperature (24 °C). Flow test validity (for fluids at all levels): smooth fluids are valid to be tested using the Flow test, while fluids that contain gristle, bone, husks, shell, fibres or lumps are not valid to be tested. Flow test (for fluids at all levels): the volume of fluid remaining in the syringe after 10 s. Fork Drip test (for fluids at level 3 and 4 in the Flow test): fluid drip and flow manner through fork prongs. Spoon Tilt test (for fluids at level 4 in the Fork Drip test): ability of fluid to hold its shape on the spoon and slide off easily when spoon is tilted.

Lubricants	Flow Test Validity		Flow Test #		Fork Drip Test *		Spoon Tilt Test ^		Final IDDSI Classification
	Texture	Interpretation	Volume Remaining (mL)	Interpretation (IDDSI Classification)	Flow/Drip/No Drip	Interpretation (IDDSI Classification)	Pass/Fail	Interpretation (IDDSI Classification)	
Heyaxon	Lumpy	Not valid	-	-	-	-	-	-	7
Magic Jelly dysphagia	Lumpy	Not valid	-	-	-	-	-	-	7
Magic Jelly adult	Lumpy	Not valid	-	-	-	-	-	-	7
Slo Tablets	Smooth	Valid	8.7	3	Flow	<3	-	-	<3
MediSpend	Smooth	Valid	9.1	3	Flow	<3	-	-	<3
Severo	Smooth	Valid	7.5	2	-	-	-	-	2
Gloup Original straw/ban	Smooth	Valid	9.9	3	Drip	3	-	-	3
Gloup Sugar Free	Smooth	Valid	9.9	3	Drip	3	-	-	3
Gloup Low Sugar	Smooth	Valid	9.9	3	Drip	3	-	-	3
Gloup Original orange	Smooth	Valid	9.9	3	Drip	3	-	-	3
Gloup Forte	Smooth	Valid	No flow	4	No drip	4	Pass	4	4
Swallow Aid	Smooth	Valid	No flow	4	No drip	4	Fail	7	7

- Test is not applicable. # Level 3: the amount remained after 10 s is no less than 8 mL, level 4: no flow. * Level < 3: flows continuously through the fork prongs. Level 3: the sample drips in slow dollops through the fork prongs. Level 4: the sample mounds above the fork. ^ Pass: contents easily slide off spoon. Fail: contents are firm and sticky.

**Table 3 pharmaceutics-12-00924-t003:** Medication lubricant classification according to IDDSI testing methods at fridge temperature (4 °C). Flow test validity (for fluids at all levels): smooth fluids are valid to be tested using the Flow test, while fluids that contain gristle, bone, husks, shell, fibres or lumps are not valid to be tested. Flow test (for fluids at all levels): the volume of fluid remaining in the syringe after 10 s. Fork Drip test (for fluids at level 3 and 4 in the Flow test): fluid drip and flow manner through fork prongs. Spoon Tilt test (for fluids at level 4 in the Fork Drip test): ability of fluid to hold its shape on the spoon and slide off easily when spoon is tilted.

Lubricants	Flow Test Validity		Flow Test #		Fork Drip Test *		Spoon Tilt Test ^		Final IDDSI Classification
	Texture	Interpretation	Volume Remaining (mL)	Interpretation (IDDSI Classification)	Flow/Drip/No Drip	Interpretation (IDDSI Classification)	Pass/Fail	Interpretation (IDDSI Classification)	
Heyaxon	Lumpy	Not valid	-	-	-	-	-	-	7
Magic Jelly dysphagia	Lumpy	Not valid	-	-	-	-	-	-	7
Magic Jelly adult	Lumpy	Not valid	-	-	-	-	-	-	7
Slo Tablets	Smooth	Valid	9.5	3	Flow	<3	-	-	<3
MediSpend	Smooth	Valid	8.9	3	Flow	<3	-	-	<3
Severo	Smooth	Valid	7.8	2	-	-	-	-	2
Gloup Original straw/ban	Smooth	Valid	9.9	3	No drip	4	Pass	4	4
Gloup Sugar Free	Smooth	Valid	No flow	4	No drip	4	Pass	4	4
Gloup Low Sugar	Smooth	Valid	No flow	4	No drip	4	Pass	4	4
Gloup Original orange	Smooth	Valid	No flow	4	No drip	4	Pass	4	4
Gloup Forte	Smooth	Valid	No flow	4	No drip	4	Pass	4	4
Swallow Aid	Smooth	Valid	No flow	4	No drip	4	Fail	7	7

- Test is not applicable. # Level 3: the amount remained after 10 s is no less than 8 mL, level 4: no flow. * Level < 3: flows continuously through the fork prongs. Level 3: the sample drips in slow dollops through the fork prongs. Level 4: the sample mounds above the fork. ^ Pass: contents easily slide off spoon. Fail: contents are firm and sticky.

**Table 4 pharmaceutics-12-00924-t004:** Classification of Gloup products according to IDDSI testing methods when poured from the bottle, measured at room (24 °C) or fridge temperature (4 °C). Flow test validity (for fluids at all levels): smooth fluids are valid to be tested using the Flow test, while fluids that contain gristle, bone, husks, shell, fibres or lumps are not valid to be tested. Flow test (for fluids at all levels): the volume of fluid remaining in the syringe after 10 s. Fork Drip test (for fluids at level 3 and 4 in the Flow test): fluid drip and flow manner through fork prongs. Spoon Tilt test (for fluids at level 4 in the Fork Drip test): ability of fluid to hold its shape on the spoon and slide off easily when spoon is tilted.

Lubricants	Flow Test Validity		Flow Test #		Fork Drip Test *		Spoon Tilt Test ^		Final IDDSI Classification
	Texture	Interpretation	Volume Remaining (mL)	Interpretation (IDDSI classification)	Flow/Drip/No Drip	Interpretation(IDDSI Classification)	Pass/Fail	Interpretation (IDDSI Classification)	
**Room temperature**									
Gloup Original straw/ban	Smooth	Valid	9.9	3	No drip	4	Pass	4	4
Gloup Sugar Free cherry	Smooth	Valid	9.9	3	No drip	4	Pass	4	4
Gloup Low Sugar raspberry	Smooth	Valid	9.9	3	No drip	4	Pass	4	4
Gloup Original orange	Smooth	Valid	No flow	4	No drip	4	Pass	4	4
Gloup Forte	Smooth	Valid	No flow	4	No drip	4	Pass	4	4
**Fridge temperature**									
Gloup Original straw/ban	Smooth	Valid	No flow	4	No drip	4	Pass	4	4
Gloup Sugar Free cherry	Smooth	Valid	No flow	4	No drip	4	Pass	4	4
Gloup Low Sugar raspberry	Smooth	Valid	No flow	4	No drip	4	Pass	4	4
Gloup Original orange	Smooth	Valid	No flow	4	No drip	4	Pass	4	4
Gloup Forte	Smooth	Valid	No flow	4	No drip	4	Pass	4	4

- Test is not applicable. # Level 3: the amount remained after 10 s is no less than 8 mL, level 4: no flow. * Level < 3: flows continuously through the fork prongs. Level 3: the sample drips in slow dollops through the fork prongs. Level 4: the sample mounds above the fork. ^ Pass: contents easily slide off spoon. Fail: contents are firm and sticky.

**Table 5 pharmaceutics-12-00924-t005:** Texture profile analysis of the medication lubricants measured at room (24 °C) and fridge temperature (4 °C). Mean values followed by unlike superscript letters (a–f) within a column are significantly different (*p* < 0.05). The data are mean ± se for three replicates.

Lubricants	Hardness (g)		Adhesiveness (mJ)		Cohesiveness		Gumminess (g)		Springiness (mm)	
	Room T	Fridge T	Room T	Fridge T	Room T	Fridge T	Room T	Fridge T	Room T	Fridge T
Heyaxon	4.38 ± 0.56 ^a^	3.67 ± 0.14 ^ab^	0.13 ± 0.00 ^a^	0.10 ± 0.03 ^a^	0.85 ± 0.01 ^ab^	0.88 ± 0.02 ^a^	3.7 ± 0.45 ^a^	3.20 ± 0.17 ^ab^	19.74 ± 0.05 ^a^	19.70 ± 0.02 ^a^
Magic Jelly dysphagia	4.17 ± 0.54 ^a^	5.17 ± 0.14 ^ac^	0.09 ± 0.01 ^a^	0.09 ± 0.03 ^a^	0.74 ± 0.02 ^ab^	0.84 ± 0.11 ^a^	3.1 ± 0.49 ^a^	4.30 ± 0.45 ^ac^	19.68 ± 0.05 ^a^	19.27 ± 0.34 ^a^
Magic Jelly adult	4.17 ± 0.36 ^a^	3.83 ± 0.49 ^ab^	0.07 ± 0.20 ^a^	0.24 ± 0.15 ^a^	0.94 ± 0.07 ^ab^	0.84 ± 0.04 ^a^	4.03 ± 0.24 ^a^	3.23 ± 0.28 ^ab^	19.16 ± 0.30 ^a^	19.64 ± 0.02 ^a^
Slo Tablets	3.67 ± 0.14 ^a^	3.17 ± 0.14 ^ab^	0.06 ± 0.01 ^a^	0.05 ± 0.01 ^a^	0.98 ± 0.02 ^a^	0.79 ± 0.05 ^a^	3.60 ± 0.08 ^a^	2.50 ± 0.24 ^ab^	19.02 ± 0.39 ^a^	14.88 ± 0.47 ^ab^
MediSpend	3.33 ± 0.36 ^a^	3.50 ± 0.00 ^ab^	0.07 ± 0.04 ^a^	0.07 ± 0.01 ^a^	0.85 ± 0.01 ^ab^	0.80 ± 0.04 ^a^	3.17 ± 0.14 ^a^	2.80 ± 0.12 ^ab^	19.37 ± 0.22 ^a^	19.08 ± 0.36 ^ab^
Severo	3.33 ± 0.14 ^a^	2.00 ± 0.24 ^b^	0.08 ± 0.02 ^a^	0.05 ± 0.01 ^a^	0.59 ± 0.13 ^b^	0.59 ± 0.09 ^a^	2.00 ± 0.50 ^a^	1.23 ± 0.33 ^a^	19.65 ± 0.04 ^a^	17.00 ± 0.43 ^ab^
Gloup Original straw/ban	4.33 ± 0.27 ^a^	5.50 ± 0.24 ^ac^	0.08 ± 0.02 ^a^	0.12 ± 0.01 ^a^	0.93 ± 0.03 ^ab^	0.80 ± 0.02 ^a^	4.03 ± 0.40 ^a^	4.4 ± 0.21 ^ac^	19.70 ± 0.06 ^a^	19.60 ± 0.19 ^a^
Gloup Sugar Free	4.67 ± 0.49 ^a^	6.50 ± 0.24 ^c^	0.09 ± 0.00 ^a^	0.13 ± 0.03 ^a^	0.83 ± 0.01 ^ab^	0.79 ± 0.02 ^a^	4.33 ± 0.10 ^a^	5.13 ± 0.23 ^bc^	16.90 ± 1.68 ^a^	18.32 ± 1.14 ^ab^
Gloup Low Sugar	4.83 ± 0.68 ^a^	6.83 ± 0.59 ^c^	0.12 ± 0.04 ^a^	0.15 ± 0.00 ^a^	0.75 ± 0.09 ^ab^	0.54 ± 0.05 ^a^	3.67 ± 0.73 ^a^	3.60 ± 0.66 ^ab^	16.79 ± 2.29 ^a^	12.55 ± 2.36 ^b^
Gloup Original orange	5.67 ± 0.27 ^a^	7.67 ± 0.14 ^c^	0.14 ± 0.02 ^a^	0.15 ± 0.01 ^a^	0.80 ± 0.04 ^ab^	0.60 ± 0.03 ^a^	4.60 ± 0.45 ^a^	4.63 ± 0.20 ^ac^	19.14 ± 0.29 ^a^	16.56 ± 1.10 ^ab^
Gloup Forte	9.17 ± 0.14 ^b^	11.00 ± 0.41 ^e^	0.33 ± 0.01 ^a^	0.36 ± 0.06 ^a^	0.83 ± 0.01 ^ab^	0.65 ± 0.05 ^a^	7.60 ± 0.19 ^b^	7.17 ± 0.43 ^c^	19.62 ± 0.06 ^a^	15.64 ± 1.67 ^ab^
Swallow Aid	16.50 ± 0.62 ^c^	18.80 ± 0.83 ^f^	1.95 ± 0.21 ^b^	1.50 ± 0.05 ^b^	0.99 ± 0.01 ^a^	1.62 ±0.15 ^b^	16.4 ± 0.41 ^c^	30.40 ± 1.37 ^d^	22.39 ± 2.02 ^a^	14.10 ± 0.58 ^ab^

**Table 6 pharmaceutics-12-00924-t006:** Extrusion test results for the medication lubricants at room (24 °C). Mean values followed by unlike superscript letters (a–g) within a column are statistically different (*p* < 0.05). The data are mean ± se for three replicates.

Lubricants	Back Extrusion				Forward Extrusion	
	Hardness (g)	Consistency (mJ)	Adhesive Force (g)	Adhesiveness (mJ)	Hardness (g)	Consistency (mJ)
Heyaxon	62.17 ± 0.82 ^ab^	7.42 ± 0.27 ^ac^	28.50 ± 2.05 ^a^	4.85 ± 0.35 ^ab^	1150.8 ± 75.0 ^a^	68.2 ± 3.92 ^ab^
Magic Jelly dysphagia	63.33 ± 9.61 ^ab^	7.67 ± 1.11 ^abe^	26.50 ± 4.14 ^ab^	4.47 ± 0.44 ^ac^	1012.5 ± 40.9 ^ab^	145.1 ± 7.88 ^c^
Magic Jelly adult	51.67 ± 0.31 ^ac^	6.33 ± 0.07 ^cde^	19.67 ± 0.76 ^ab^	3.39 ± 0.01 ^bc^	936.0 ± 31.5 ^abc^	133.5 ± 6.18 ^c^
Slo Tablets	32.50 ± 0.20 ^c^	4.31 ± 0.01 ^d^	16.83 ± 0.14 ^b^	2.87 ± 0.02 ^c^	131.3 ± 1.7 ^d^	17.7 ± 0.22 ^e^
Medispend	36.67 ± 0.12 ^c^	5.00 ± 0.02 ^cd^	18.50 ± 0.24 ^ab^	3.32 ± 0.06 ^bc^	315.5 ± 31.1 ^ed^	22.7 ± 2.84 ^de^
Severo	40.17 ± 0.85 ^ac^	5.46 ± 0.08 ^cde^	22.83 ± 0.27 ^ab^	3.57 ± 0.09 ^bc^	405.2 ± 46.9 ^def^	42.8 ± 0.91 ^ae^
Gloup Original straw/ban	80.67 ± 1.55 ^b^	10.15 ± 0.15 ^b^	43.67 ± 0.54 ^c^	6.04 ± 0.10 ^a^	681.8 ± 11.9 ^bf^	86.1 ± 0.2 ^b^
Gloup Sugar Free	78.33 ± 1.83 ^b^	9.82 ± 0.12 ^ab^	41.50 ± 0.62 ^c^	5.71 ± 0.13 ^a^	472.5 ± 76.6 ^def^	50.4 ± 2.71 ^acf^
Gloup Low Sugar	80.83 ± 1.33 ^b^	9.87 ± 0.20 ^ab^	40.50 ± 1.65 ^c^	5.51 ± 0.09 ^a^	642.6 ± 70.0 ^cef^	68.4 ± 3.31 ^ab^
Gloup Original orange	106.50 ± 1.95 ^d^	13.57 ± 0.38 ^f^	59.67 ± 1.77 ^d^	7.84 ± 0.28 ^d^	665.3 ± 32.5 ^be^	77.7 ± 7.34 ^bf^
Gloup Forte	219.50 ± 1.89 ^e^	28.11 ± 0.40 ^g^	77.00 ± 0.85 ^e^	21.56 ± 0.56 ^e^	1258.5 ± 25.8 ^a^	164.8 ± 2.34 ^c^

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
