# Peer review of "Are Medication Swallowing Lubricants Suitable for Use in Dysphagia? Consistency, Viscosity, Texture, and Application of the International Dysphagia Diet Standardization Initiative (IDDSI) Framework"

_pharmaceutics, 2020, doi:10.3390/pharmaceutics12100924_

Round 1

Reviewer 1 Report

General Comments:

The paper is very well-written, clear and specific. The subject of investigation is novel, timely and important. The experiments have been rigorously conducted and described.

There is one major limitation to the applicability of the work which really stands out as it is central to the purpose of the study: These liquids/gels may be suitable for a person with dysphagia on their own, but no mention is made of their suitability as they are intended to be used: i.e. a bolus containing the lubricant plus a solid oral medication form. Admittedly this is not a straightforward question to resolve, in-vitro or in-vivo, however the issue is so fundamental I think it deserves high priority throughout the paper. The lubricants themselves are not intended to be consumed on their own (as tested here) without a medication included. In the discussion, the authors mention (line 509) that “…products containing a mixed texture are generally viewed as inappropriate for use by people with dysphagia…” but here they are only referring to the soft lumps within the lubricant material, not a capsule or tablet.

Specific Comments:

Abstract: I liked how the main outcomes are included within the abstract, helping readers.

Line 188 Bostwick consistometer, it might be interesting to state the volume of liquid used for each test. Often this is 75ml, much more than the volume used in practice.

Line 270 & 494 – where not possible to achieve a yield stress reading, is this because the materials did not exhibit a measurably large yield stress effect? Or any other reason or characteristic?

Table 3 (and 4, and descriptions elsewhere within text) – The IDDSI classification of “< 3” might be simply interpreted as “2” -is this incorrect? If there is a difference, it might benefit from further clarification. Perhaps this consistency isn’t a direct match to any IDDSI level?

Figure 1 – it would help to separate panels A & B: their shared x-axis makes the values a little tricky to read, particularly since A has a broken axis too.

Line 338 & Table 5 – When you highlight significant differences between products & parameters, it might be more illustrative to include a comparison of the value magnitudes, e.g. “approximately twice as hard” if appropriate. This is with the aim of differentiating between statistical significance and practical significance, and making the large tables of data more readily accessible to the readers.

Line 239 and 452 – it reads as slightly odd that a product was classified as Level 2 because the flow test measure was outside Level 3, rather than saying that the measure was within the definition for Level 2 (4-8ml remaining).

The different measurement techniques clearly assess slightly different parameters of the products which don’t always align between products: It might be helpful to introduce this concept to the reader near the start of the discussion, to potentially avoid confusion when discussing the results.

You don’t mention refrigerating the Botwick device when measuring chilled products – I’m guessing this wasn’t deemed necessary or conventional, but might have affected the results. No change necessary, just to consider.

479 – rather than describe an “increase in viscosity” after pouring it is more likely that the pumping shear had decreased the viscosity and that effect didn’t occur when the material was poured. NB it is very commendable that you investigated this effect.

Reviewer 2 Report

Paper is interesting, it is well writen

Reviewer 3 Report

Dysphagia describes problems with swallowing within oral, pharyngeal or oesophagael phases. It is not just about aspiration but about making the bolus an appropriate size and lubricating it, ensuring that the bolus does not cause choking or aspiration, ensuring that the bolus can pass through a narrowed oesophagua. The introduction swaps between choking and aspiration when a lubricant is designed to prevent choking but if it has the incorrect texture can itself cause aspiration.  This clarity and distinction is not currently provided. 

The initial background requires reordering to make the message clearer for the reader.  Definition, prevalence, cause, description of swallow and problems associated with different stages, assessment and guidance regarding safe swallow.  Medication specific problems and role of lubricants. 

The aim which is 'to consider... safety' seems vague and incorrect.  It is to describe the rheological properties of lubricants to inform selection and ultimately enhance safety.  There is no assessment of safety.  Similarly suitability depends on the nature of the dysphagia and IDDSI requirement, so again it is about identifying the most suitable for people requiring texture modification.

The text after the aim is actual methods and should be move to that section

After reading the background I am unclear as to how the different properties measured, other than IDDSI, can be used to inform actual practice or why PH was measured?

Table 1 is a result with methods in the title? Seems unconventional to locate this information within this section.

The statistical analysis is not related to the aim or any stated objectives and as presented feels like an add on or a trawling exercise as no rationale is provided in the background for doing these comparisons.

The results include data on fridge and room temperature storage, which again I could not see a rationale or explanation for.

Table titles are so long they are unreadable and require shortening with the additional information provided in the methods

Table 4 focusses on Gloop only which seems to be more tests and results with no rationale

What are the plus and minus values in tables 5 & 6. I suspect SE and therefore we dont really need lots of p values as the reader can see what the number is and whether it overlaps with other results.

Start of discussion is actually background. Does not discuss the results

Discussion repeats data from the results which is not required

Discussion would benefit from main messages, limitations then detailed discussion, perhaps in order of product.  Currently it is difficult to read and draw out what has been learned from this or what the data means in practice.

Within the conclusion the authors state that the lubricants may not be safe for dysphagia. I think the authors means those with aspiration and requiring a bolus which will hold its form.  The language needs to be more precise.

I suspect that these lubricants are for individuals who are at risk of choking or psychologically adverse to taking solid dose forms due to anticipation of choking (usually due to oral phase dysfunction), rather than those at risk of aspiration (Pharyngeal phase dysfunction). Consequently for the majority of patients for whom these products are targetted the texture is irrelevant providing it lubricates the tablet or capsule.

The authors swap between tablet/capsule and pill but the latter is very old terminology for sold dose formulations and should no longer be being used in scientific journals.

In summary, the paper does not make a coherent arguement for the research, does not provide a good rationale for much of what was done, undertakes statistical tests which are largely unecessary, includes sections of text and data in the wrong sections and provides a discussion which is hard to read and follow, not adhering to more usual conventions.

The authors need to decide what the aim and objectives of the paper are, prepare a background which justifies them, provide results which are focussed on meeting the objectives and a discussion which follows a more conventional order.

Paper would benefit from greater focus and not attempting to present so much data in one place.  This would probably benefit from splitting into two papers.

Reviewer 4 Report

The design and development of patient-centric formulations are particularly relevant in the case of dysphagic problem. in this context, the manuscript focused on characterizing the technological properties of medication lubricant to aid swallowing of solid dosage forms. The work resulted very interesting and should be considered for the publication after some minor revisions. 

In particular:

  • It could be interesting for the readers to have a look on about the classification of the tested product (e.g., medical devices, medicinal product) and if they are available only in the Australian market or not;
  • The IDDSI framework is not detailed in the text. Some tips are reported in the footnote of the tables. however, for improving the readability of the results, the authors should add a table with all the criteria scales
  • The way used for reported the temperature conditions in the text and the table should be harmonized preferring the use of 4/24°C and not room/fridge t). 
  • Starting from the experimental results, the authors are invited to review the discussion to try proposing a sort of the "specification range" for selecting the medication lubricant.

Round 2

Reviewer 3 Report

Happy with responses to my comments